# Post-Treatment Status of Impacted Maxillary Central Incisors following Surgical-Orthodontic Treatment: A Systematic Review

**DOI:** 10.3390/medicina57080783

**Published:** 2021-07-30

**Authors:** Alvyda Žarovienė, Dominyka Grinkevičienė, Giedrė Trakinienė, Dalia Smailienė

**Affiliations:** Department of Orthodontics, Medical Academy, Lithuanian University of Health Sciences, Lukšos-Daumanto 6, LT-50106 Kaunas, Lithuania; dominyka.narb@gmail.com (D.G.); gyd_trakiniene@yahoo.com (G.T.); dsmailiene@gmail.com (D.S.)

**Keywords:** incisor, maxillary, impacted, surgical-orthodontic treatment, periodontal status

## Abstract

*Background and Objectives:* The aim of this systematic review was to summarize currently available data of published articles that have investigated the post-treatment status of impacted maxillary central incisors (ICI) treated by the surgical-orthodontic approach. *Materials and Methods:* MEDLINE, Scopus, Cochrane Oral Health Group’s Trials Register, and ScienceDirect electronic databases were systematically searched with no publication date restrictions up to January 2021. Data assessing the status of ICI after combined surgical-orthodontic treatment and forced eruption duration were extracted, and the quality of the studies was evaluated. *Results:* In total, 7074 studies were identified, of which 42 articles were assessed for eligibility through full-text evaluation. Seven included studies (five retrospective studies, one randomized clinical trial, and one prospective clinical trial) met the inclusion criteria, representing 211 patients with unilaterally impacted maxillary incisors. The risk of bias ranged from moderate to high. The results show that the root length of immature ICIs increased significantly but remained shorter than that of homonym teeth at post-treatment. Periodontal parameters of treated ICIs were in a clinically acceptable range. Measurements of the alveolar bone showed a reduction of bone thickness and support. The average forced eruption duration ranged from 8.0 ± 4.5 to 14.41 ± 4.03 months. *Conclusions:* Based on existing evidence, it is reasonable to conclude that the surgical-orthodontic treatment affected the post-treatment status of ICI; however, the current literature is insufficient to draw concrete conclusions. Further well-conducted multi-center randomized studies with a large sample are needed to confirm this statement.

## 1. Introduction

An impacted tooth is defined as a tooth that fails to reach the occlusal plane after the normal age of eruption or when its contralateral tooth has already erupted for at least six months with a fully developed root. Previous studies reported that the impaction of maxillary central incisors occurs with a prevalence of 0.03–0.2% [1] In the retrospective study conducted by Tan et al. [2], the incidence of impacted incisors was 2.0% among the Chinese orthodontic patient population, and maxillary central incisors were the most frequently affected teeth (70.6% of all impacted incisors).

The etiology of eruption failure is multifactorial. The etiological factors for the delayed eruption of maxillary central incisors can be classified into two main categories, namely local and systemic. Local factors include dilacerations, supernumerary teeth, odontoma, dental trauma, the ectopic position of tooth bud, dentigerous cyst, lack of space in the arch, premature loss of deciduous teeth, cleft lip and palate, and tissue scar created as a result of early tooth extraction [2,3,4,5].

Regardless of their low incidence, impacted maxillary central incisors (ICI) constitute a huge aesthetic, developmental, functional, and psychological problem. Impaction leads to serious complications, such as the ectopic position of the unerupted permanent tooth itself and adjacent teeth, incisor transposition, space loss and midline shift, irregular dentition, and root resorption of adjacent teeth [1]. Because of its location in the dental arch, the central incisor plays an important role in facial esthetics and phonetics, therefore its normal position and morphology are crucial. A diagnosis of an impacted permanent incisor is often directly followed by an attempt to reposition the impacted tooth into the dental arch. Common approaches for the management of impacted maxillary permanent incisors include early interceptive measures to facilitate the eruption of displaced maxillary incisors or surgical exposure of the tooth’s crown with a subsequent orthodontic alignment of the tooth.

According to the literature, after removal of the cause of impaction (removal of supernumerary teeth and expansion of the dental arch), most impacted maxillary central incisors erupt spontaneously within 1–2 years, and this is more observed in younger patients [6,7,8,9,10]. The recorded rate of spontaneous eruption ranges from 30.3% to 89.4% of cases in different studies and depends on the initial maturation of the root of the impacted tooth, initial vertical position, the degree of the angulation of the impacted incisor, the form of the obstacle, and the additional orthodontic expansion of the dental arch [6,7,8,9].

However, some impacted incisors still do not erupt and require additional treatment by the surgical exposure of the impacted tooth, followed by orthodontic traction. Some authors demonstrated relatively good treatment results following this technique [11,12,13], while others reported less successful treatment outcomes [14].

Several studies with various patient samples have studied different treatment protocols and presented different conclusions about surgical-orthodontic treatment outcomes [11,12,14,15,16,17,18,19,20,21,22,23]. There is no summarized evidence about the outcomes of the surgical-orthodontic treatment approach for impacted maxillary incisors, and there have not been previous systematic reviews on this subject. Therefore, the aim of this systematic review was to assess the methodological quality and summarize currently available data of published articles that have investigated the post-treatment status of impacted maxillary central incisors treated by the surgical-orthodontic approach.

## 2. Materials and Methods

This systematic review was conducted according to the PRISMA statement (Preferred Reporting Items for Systematic Reviews and Meta-Analyses) [24], and the protocol was registered with PROSPERO (registration number of CRD42021211486).

### 2.1. Eligibility Criteria

According to the Participants Intervention Comparison Outcome Study design schema (PICOS), the study included randomized, prospective, and retrospective controlled non-randomized trials (S) on human patients of any age or sex with unilaterally impacted maxillary permanent central incisor (P) comparing the outcome of surgical-orthodontic treatment (I) versus a normally erupted contralateral incisor (C). The primary outcomes (O) of this systematic review included post-treatment periodontal, alveolar bone, and root conditions. The secondary outcome was the average forced eruption duration.

Exclusion criteria were case reports and series, literature reviews, studies on animals, studies on patients with genetic syndromes, and severe facial malformations.

### 2.2. Information Sources and Search Strategy

MEDLINE (via Ovid and PubMed), Scopus, Cochrane Oral Health Group’s Trials Register, and ScienceDirect electronic databases were systematically searched by two authors up to January 2021 with no publication date restrictions.

A list of both Mesh and non-Mesh terms was drawn up to restrict the research field to articles that were related to the study subject. The search strategy for MEDLINE (via Ovid and PubMed) was the following: ((“Incisor” [All Fields] AND “maxillary” [All Fields]) OR (“Incisor” [All Fields] AND “upper” [All Fields])) AND (“Impacted” [All Fields] OR “retained” [All Fields] OR “ectopia” [All Fields] OR “eruption” [All Fields] OR “displaced” [All Fields] OR “malpositioned” [All Fields]). The search strategy was appropriately adapted for Scopus, Cochrane Oral Health Group’s Trials Register, and ScienceDirect electronic databases. The limitations were applied as follows: articles in English. In addition, the reference/citation lists of included trials were manually searched for any additional trials.

### 2.3. Study Selection

Before beginning the search in the selected databases, the search strategy was discussed among three investigators. The study selection was then performed independently by two researchers. Selection and filtration were done by the assessment of the titles of the articles and their abstracts, and duplicates were removed. If the article corresponded to the criteria of inclusion of review, then the entire article was read to make the final decision. Disagreements were resolved by consensus between the two reviewers, and a third author was consulted when necessary.

### 2.4. Data Extraction and Management

Characteristics and data of included studies that were considered eligible were extracted independently by two reviewers. To record the desired information, a data extraction form was developed (based on the Cochrane Consumers and Communication Review Group’s data extraction template), pilot-tested on five randomly selected included studies, and refined accordingly. In the event of any disagreement, a third reviewer was involved. The following variables were recorded for each reviewed article: author, year published, type of study (retrospective, prospective, controlled, or not controlled), characteristics of study participants (sample size, age, and sex), inclusion and exclusion criteria, intervention used (the type of surgery and the type of orthodontic mechanics), used evaluation method (radiographs and clinical examination), treatment outcomes: post-treatment periodontal parameters (probing depth (PD), gingival recession, and contour), root measurements (root length (RL) and resorption (RR)), alveolar bone measurements (bone level and the thickness of alveolar bone), and treatment duration, defined as the time between applying the initial traction to the time of good alignment of the impacted incisor in the dental arch and follow-up period.

### 2.5. Quality Assessment

To assess the quality of studies, the ROBINS-I tool for non-randomized and the Cochrane risk of bias tool for randomized studies were used. Two reviewers evaluated studies individually, and any disagreements between reviewers were solved by discussion to reach an agreement.

## 3. Results

### 3.1. Study Selection

The protocol for the present review followed the guidelines presented in the PRISMA statement (Figure 1). In total, 7074 articles were initially identified in the electronic databases, of which 3647 were found to be duplicates. Then, 3427 articles were screened based on their title and abstract, of which 3385 records were excluded because they were not related to the subject or did not fulfill the eligibility criteria. The final 42 articles were assessed for eligibility through full-text evaluation, after which 35 were excluded. The seven articles that met the inclusion criteria were included in the qualitative synthesis [11,12,14,16,18,19,20].

### 3.2. Study Characteristics

The characteristics of the included studies are summarized in Table 1. Five of the included studies were retrospective, one randomized clinical trial, and one prospective clinical trial.

Participants. Studies included a total of 211 patients with successfully treated unilaterally impacted maxillary central incisors. There were variations in the total sample size (from 12 to 80 patients) and age (from 6.25 to 38 years) with male patients being 47.87% (101/211 patients) of the studies included. Four studies included only patients in the mixed dentition period [12,18,19,20] and three studies had a very wide distribution of patient’s ages [11,14,16]. In two studies, the treatment group consisted of inversely impacted maxillary central incisors with few cases of dilacerated incisors [12,19].

Intervention. In all seven studies, the impacted maxillary incisor was correctly positioned in the dental arch through orthodontic treatment and forced eruption. In five studies, the closed-eruption surgical technique method was used [11,12,16,18,20]. Two studies also included incisors exposed through the apically repositioned flap or open-eruption techniques [14,16] and one study did not report surgical exposure technique [19].

Six of these studies analyzed different types of radiographs, such as cone beam computed tomography (CBCT), periapical radiographs, lateral cephalograms, and panoramic radiographs [11,12,14,16,18,19,20]. Three studies included a periodontal clinical examination [11,14,20].

### 3.3. Quality Assessment

Only one RCT [17] was included in this systematic review, and this trial was of “high” overall risk of bias due to blinding of the reported results, unclear information of allocation sequence concealment, and selective reporting (Table 2).

The risk of bias within non-randomized studies for the two trials was evaluated to have an overall “moderate” bias due to certain discrepancies in confounding, selection, intended interventions, and measurement of outcomes domains [12,19]. Three non-randomized studies [11,14,18] were found to have an overall serious risk of bias and one study [16] presented a critical risk of bias. The most problematic domains were associated with a lack of blinding, assessment of confounding, and outcomes measurement (Table 2).

### 3.4. Results of Individual Studies

The results of the included studies are summarized in Table 3 and Table 4.

Four of the included studies analyzed the maxillary central incisors root morphology and development after the treatment [12,16,18,19]. Three CBCT studies investigated the root development of ICI, whose root was immature at the beginning of the treatment [12,18,19]. Shi et al. [18] stated that the root length (RL) of impacted incisors increased significantly during the treatment period from 6.67 ± 1.94 to 10.66 ± 2.10 mm and the difference in root length of the ICI and CCI at post-treatment was not significant (*p* = 0.771), although before treatment the root length of ICC was significantly shorter. Hu et al. [12] investigated root length at the follow-up and reported a significant increase in both treated ICI and CCI roots length post-treatment by 1.36 (1.05)–2.36 (1.12) mm. However, in terms of dilacerated roots, direct RL remained significantly shorter. Sun et al. [19] and Ho and Liao [16] presented conclusions that the RL of impacted incisors was significantly shorter than that of homonym teeth at post-treatment. According to results, the mean RL of treated impacted incisors was 10.9 (3.5) in comparison to 13.7 (2.9) of CCI (*p* < 0.001) [16]. Ho and Liao [16] analyzed the root resorption (RR) during surgical-orthodontic treatment on periapical radiographs and revealed that extruded incisors had significantly greater RR than the naturally erupted incisors. In addition, the RR was significantly associated with initial crown height and depth, treatment duration, and root dilaceration.

Three studies evaluated the status of periodontal parameters of treated ICIs [8,11,17]. In the study by Sfeir et al. [20], the average periodontal probing depth (PD) was in a clinically acceptable range after the treatment of ICI during the mixed dentition period (range 2.46 (0.24)–2.74 (0.25) mm), although was deeper than PD of CCI. When comparing Discontinuous and Continuous traction groups, only mesial PD showed a statistically significant difference and was deeper in the Continuous traction group. The conclusions made by another group of researchers showed deeper PD post-treatment [11,14], and the mean probing depth of ICI and the CCI also differed significantly; this was especially true for the disto-labial, palatal [11], and mesio-labial areas [14]. Becker et al. [11] showed statistically significant deeper mean probing depth of treated ICC (2.46 mm (SD not indicated)) than CCI (2.25 mm). Chaushu et al. [14] also showed significantly deeper mean PD of treated ICC (2.35 mm (SD not indicated)) in comparison to CCI values (2.09 mm).

Concerning gingival status, the results of studies show significant post-treatment reduction in the width of the attached gingiva by 0.28-1.06 mm and an increased number of cases with irregular gingival contour by 33–72.7% for ICI treated with the open eruption technique [11,14].

Four of the included studies investigated the alveolar bone condition in treated impacted incisors by using CBCT [12,18,19,20] and two studies using periapical radiographs [11,14]. Evaluating records of patients for whom treatment was accomplished during mixed dentition, labial bone thickness at the crest and the apex of ICI and CCI were significantly thinner after treatment than the corresponding lingual values [12,18]. Shi et al. [18] showed an alveolar bone loss for ICI on the labial side by 29.34% (18.7%) and on the lingual side by 9.48% (12.10%) (*p* < 0.05). For CCI, the result of bone loss between labial and lingual sides also was significant (13.02% (8.64%) bone loss on labial and 6.97% (5.97%) on lingual sides). Besides, alveolar bone loss for ICI on the labial side was statistically significantly greater than for CCI. However, during the 2-year follow-up period, the labial bone thickness at the apex of the impacted incisors increased significantly [12]. When evaluated alveolar bone height loss, Sfeir et al. [20] found a bone height loss on labial and palatal aspects, although it was significantly smaller when orthodontic traction was discontinued for a month after crown emergence compared to continuous orthodontic traction technique. Sun et al. [19] showed alveolar bone loss of 2.72 (1.32) mm on the labial side and 2.02 (1.32) mm on the palatal side for ICC. In three of the studies [12,18,20], the surgical technique was CET; in the study by Sun et al. [19], the method of surgery was not specified. The findings from the studies with older patients show the reduction of bone support on mesial [11,14] and distal [11] aspects of ICI post-treatment on periapical radiographs. This result was observed in both closed and open eruption surgical techniques’ treatment groups.

In four studies, the *secondary* outcome assessed was an average forced eruption duration, which ranged from 8.0 ± 4.5 to 14.41 ± 4.03 months [12,14,16,18].

## 4. Discussion

### 4.1. Root Morphology and Development

Shi et al. [18] and Hu et al. [12] confirmed that impacted incisors had continuous and similar growth as did the contralateral incisors. In a recent systematic review and meta-analysis evaluating spontaneous eruption of impacted maxillary incisors, a clinical recommendation was made to wait for the eruption of the tooth for a period of 12–36 months after surgical removal of the obstacle impeding the eruption of a maxillary anterior permanent tooth [25]. The calculated average eruption potential of impacted anterior maxillary teeth following such procedure was approximately 65.5%, with a higher odds ratio for patients under 9 years of age. However, the present analysis shows that the root of the impacted incisor could achieve better development if treated early [19]. Early treatment of teeth with immature roots could free the root apices from the restrictions for growth from the adjacent anatomical structures and allow expressing the full growth potential. Therefore, a long waiting time of 2–3 years for self-correction is inexpedient from the root formation point of view, especially for older patients.

Concerning root resorption, Ho and Liao [16] found that extruded incisors had significantly greater root resorption than the naturally erupted incisors. However, the age of patients in this study ranged from 6.4 to 20.6 years, therefore it is impossible to distinguish whether the shorter root could have been due to RR or due to disturbances during formation. Despite that, the authors stated that RR was not associated with the patient’s age and was significantly associated with initial impaction depth, treatment duration, and root dilaceration.

Tooth root dilaceration is an especially important factor in both the etiology of tooth impaction and treatment planning. This condition is a primary obstacle to successful treatment, which makes the extrusion complicated and can lead to the need for multiple surgeries or even extraction of the impacted dilacerated incisor. Dilaceration of impacted maxillary incisors may develop due to acute mechanical injury to the primary predecessor tooth or ectopic development of the tooth germ [4]. Stewart [26] studied the phenomenon of incisors dilaceration and found that only 22% of cases were due to injury, while other cases arise due to the ectopic location of tooth buds and the presence of supernumerary teeth and cysts. The roots of inversely impacted maxillary central incisors continue developing but their potential is limited. Chaushu et al. [21] reported that the failure of ICI surgical-orthodontic treatment was most often associated with root dilacerations and subsequent ankylosis. Summarizing data from studies, ICI root dilaceration was significantly associated with root resorption [16] and more than 3 months longer extrusion time in comparison to impactions caused by obstruction [21].

### 4.2. Periodontal Status

The overall results show acceptable periodontal status after treatment of impacted maxillary central incisors. Only one study evaluated periodontal condition after the treatment of ICI during the mixed dentition period and showed clinically acceptable results, especially in the Discontinuous traction group [20]. Studies with adult patients reported the average PD, although significantly deeper on treated ICI than on CCI, was in a clinically acceptable limit when average PD did not exceed 3 mm (11,14). Results of the study by Farronato et al. [17] with adult patients also indicate good periodontal status with an average PD of 1.69 ± 0.25 mm at the end of treatment. They found that the mean clinical attachment level, probing depth, and soft tissue recession values increased significantly during the 1-year follow-up period when compared to the immediate post-treatment. The results of included studies show that deeper PD tended to form in certain areas of treated ICI; however, these areas were quite different [11,14,20]. This may be because the research samples were small and did not have sufficient power to draw strong conclusions.

### 4.3. Alveolar Bone Condition

The radiographic findings confirm the results of the clinical investigation: there was 5–10% less bone support on treated ICI than on normally erupted incisors [11,14] and a significant reduction of the labial bone thickness of the impacted incisors immediately after treatment [12,18]. Chaushu et al. [14] concluded that such a result was statistically and clinically significant. Importantly, significant alveolar bone reduction on the labial side was also observed for CCI [12,18]. It may be assumed that specific initial labially inclined position of impacted incisors, and with it associated surgical intervention and orthodontic mechanics involving tipping movement are more vulnerable for the labial side of the alveolar bone during treatment. Such a character of bone loss may explain the increased number of cases with irregular gingival contour after treatment of ICI [11,14]. Nevertheless, Sun et al. [19] stated that early treatment of impacted incisors could reduce the risk of alveolar bone loss on the labial side. Besides, based on the results, the alveolar bone could be able to gain a certain amount of regeneration [12].

Concerning the post-treatment status of the attached gingiva [14], shape of gingival contour [14], and periodontal parameters [11,14], the CET showed a superior outcome when compared with OET. Comparing CET and OET, Chaushu et al. [15] concluded that the choice of surgical technique had important implications to periodontal support and appearance. According to the localization, labially impacted maxillary incisors can be compared to labially impacted canines. Incerti-Parenti et al. [27] compared different surgical techniques in a systematic review and found that both techniques (open eruption and an apically positioned flap technique) had periodontal outcomes comparable with the control group, but none of the included studies evaluated periodontal status after closed eruption technique with the control group. Lee et al. [28] in a split-mouth comparison found that the closed eruption technique affected the periodontal recession of treated labially impacted maxillary canines, even though the effect was clinically insufficient.

All included studies were of the split-mouth design. This might distort the final clinical and radiological results. Therefore, two investigated studies also included periodontal evaluation of lateral incisors and did not find any statistically significant difference between the adjacent and contralateral lateral incisors PD [11,14]. By contrast, the attached gingiva of the adjacent to ICI lateral incisor was significantly smaller than for the contralateral one, while the crown length was significantly greater [11].

### 4.4. The Average Forced Eruption Duration

In studies with immature ICI, it ranged from 10.16 ± 2.73 to 14.41 ± 4.03 months, but, in studies with older patients, it was shorter (from 8 to 10 months). Other studies’ results show that the treatment time needed for alignment of the unerupted incisors was significantly correlated to the patient’s age [21,22,23]. Chaushu et al. [21] found that the duration of the total treatment and each stage of treatment was longer in the older patient group; however, the differences did not reach statistical significance for the tooth forced eruption stage. Other factors resulting in longer treatment were impaction height [22,23], incisor length [22], and bilateral impaction [21]. However, conflicting results were found for tooth angulation [7,21,23]. In addition, when comparing different treatment techniques, Lygidakis et al. [23] concluded that the best treatment time results were achieved when treatment started from space creation, followed by surgical exposure and orthodontic traction, or waiting for spontaneous eruption, while the worst results were shown when surgical exposure was performed without creating a space.

### 4.5. Limitations

Most included articles were retrospective cohort studies with one prospective and one randomized clinical trial, and the quality was mainly medium. It means that the results of these studies should be interpreted with caution. The main limitations of included articles were blinding, assessment of confounding, nonhomogeneous study designs, and small sample sizes. Due to the low prevalence of the anomaly, it is difficult to collect study groups of sufficient size. The other limitation of the analysis was that the patient population was not the same across studies. The age of the patients and the research methods differed significantly, which makes meta-analysis impossible.

## 5. Conclusions

ICI treated by the surgical-orthodontic approach shows a slightly shorter root length with potential for continuous root growth if treated early. There is a significant reduction of vertical bone support and labial bone thickness of ICI after treatment. However, further well-conducted multi-center randomized studies with a large sample are needed to confirm this statement.

## Figures and Tables

**Figure 1 medicina-57-00783-f001:**
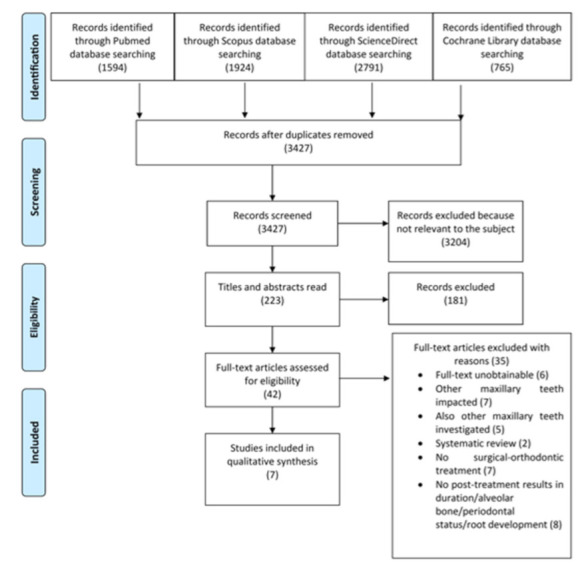
PRISMA flow diagram.

**Table 1 medicina-57-00783-t001:** Summary of the characteristics of included studies.

Authors	Study Design	Study Sample: Patients (M/F); Mean Age (Years) at T0; No. of ICI; Groups	Intervention: Type of Surgery, Type of Traction	Methods and Timing of Evaluation and Mean Follow-Up Period	Eligible Outcome
Becker et al. [11]	RS	21 (6/15); 17.5; The treatment group—No. of ICI: 21 The control group—21 normally erupted CCI.	CET Ortho-T—light traction, maintained on the ligature wire	-Periodontal clinical examination (T2) -Periapical X-rays (T2) Follow-up—4.5 years	Probing depth (PD) Width of the attached gingiva Gingival contour Bone support
Chaushu et al. [14]	RS	12 (4/8); 22 (from 15 to 38); The treatment group—No. of ICI: 12 The control group—12 normally erupted CCI.	OET Ortho-T—light traction, maintained on the ligature wire, initiated shortly after surgery.	-Periodontal clinical examination (T2) -Periapical X-rays (T2) Follow-up—10.5 months (range 3–25 months)	Average forced eruption duration Probing depth (PD) Width of the attached gingiva Gingival contour Bone support
Ho and Liao, [16]	RS	80 (32/48); 9.2 ± 2.3 (from 6.4 to 20.6); No. of ICI: 80 ICI, no mechanical obstacle to eruption. No dilaceration group (n = 64). Root dilaceration group (n = 16). The control group—80 normally erupted CCI.	CET—for deeply ICI (*n* = 29) OET—for labially and not far apically ICI (*n* = 51) Ortho-T—initiated 1–2 weeks after surgery with a bonded orthodontic traction device (force approximately 100 g.)	Cephalometric (T0), Periapical radiographs (T1). Follow-up—21.8 ± 28.6 months (range 6.0–146.1 months)	Average forced eruption duration Root length (T1)
Shi et al. [18]	RS	30 (20/10); 8.44 ± 1.20 (from 6.5 to 11.2); The treatment group—No. of ICI: 30 immature ICI. Root dilacerations *n* = 24. The control group—30 naturally erupted immature CCI.	CET Ortho-T—elastomeric chain, force 100 *g*. 19 patients had an orthodontic reopening of incisor space before surgical treatment.	CBCT (T0 and T2). Follow-up—1.76 ± 3.41 (range 0–15 months)	Average forced eruption duration Root length Alveolar bone loss on labial/lingual sides The alveolar bone thickness of labial/lingual alveolar crest Lingual/labial alveolar bone thickness at the root apex
Sun et al. [19]	PCT	28 (13/15); 8.2; No. of ICI: 28 labial inversely ICI in the mixed dentition. Early-treated group (ET, n = 14; root formation stages 7 and 8) Later-treated group (LT, n = 14; root formation stages 9 and 10) The control group—28 CCI with normal root formation and orientation.	SE (modification not described) and Ortho-T	CBCT (T1) Follow-up—not defined	Root length Alveolar bone loss in lingual/labial sides Alveolar bone thickness in lingual/labial sides
Hu et al. [12]	RS	12 (5/7); 7.80 ± 0.91 (from 6.25 to 9.42); The treatment group—No. of ICI: 12 labial inversely ICI, immature root at the beginning of treatment The control group—12 normally erupted CCI.	CET Ortho-T—using the Guide rod appliance	CBCT (T1, T2) Follow-up—24.57 ± 4.33 months (range 18.40–34.53 months).	Average forced eruption duration Root length: dilacerated root length (DRL) and directed root length (DIRL) Labial/lingual alveolar bone vertical loss Labial/lingual alveolar bone thickness of the crest Labial/lingual alveolar bone thickness at the apex
Sfeir et al. [20]	RCT	28 (21/7); Age ranged from 8 to 10 years; No. of ICI: 28 ICI, due to an obstacle, insufficient space, or insufficient eruption potential. Discontinuous traction group (DT, n = 13) Continuous traction group (CT, n = 15) The control group—28 completely erupted CCI	CET DT—traction interruption for a month after crown emergence CT—continuous traction. (elastomeric chain, the force of 1 ounce (30 gm))	-Periapical digital X-rays -anterosuperior CBCT scans -periodontal probing (T2) Follow-up—not defined	Mesial and distal alveolar ridges of the ICI and CCI (MBL, DBL) Labial and palatal alveolar ridge levels of ICI and CCI (LBL, PBL) Periodontal probing of ICI and CCI on all four sides of each tooth (MP, LP, DP, PP)

RS, Retrospective study; RCT, Randomized clinical trial; PCT, Prospective controlled clinical trial; M, males; F, females; ICI, maxillary impacted central incisors; CCI, contralateral central incisors; SE, surgical exposure; CET, closed-eruption surgical technique; APFT, apically positioned flap technique; OET, open-eruption surgical technique; DT, discontinuous traction; CT, continuous traction; Ortho-T, orthodontic traction; T0, Before treatment; T1, Immediately after treatment; T2, At the follow-up.

**Table 2 medicina-57-00783-t002:** Summary of risk of bias assessment for randomized study (Cochrane risk-of-bias tool) and risk of bias assessment for non-randomized study (the ROBINS-I tool).

**The Cochrane Risk of Bias Tool for Randomized Studies**								
**Study**	**Random Sequence Generation**	**Allocation Sequence Concealment**	**Blinding of Participants and Personnel**	**Blinding of Outcome Assessment**	**Incomplete Outcome Data**	**Selective Reporting**	**Other Potential Bias**	**Overall**
Sfeir et al. [20]	Low	Unclear	High	High	Low	Unclear	Low	High
**The ROBINS-I Tool for Non-Randomized Studies**								
**Studies**	**Confounding**	**Selection Bias**	**Classification of** **Interventions**	**Intended** **Interventions**	**Missing Data**	**Measurement of Outcomes**	**Reported Result**	**Overall**
Becker et al. [11]	Serious (difference in age, sex distribution, and follow-up time).	Moderate	Low	Serious (missing information about treatment details)	Low	Moderate (not blinded assessor)	Low	Serious
Chaushu et al. [14]	Serious (difference in age, sex distribution and follow up time)	Moderate	Low	Serious (missing information about treatment details)	Low	Moderate (not blinded assessor)	Low	Serious
Ho and Liao, [16]	Critical (difference in age)	Moderate	Low	Low	Low	Moderate (not blinded assessor)	Low	Critical
Shi et al. [18]	Moderate	Moderate	Low	Low	Low	Serious (no method error, not blinded assessor)	Low	Serious
Sun et al. [19]	Moderate	Moderate	Low	Moderate (treatment details partially provided)	Low	Moderate (not blinded assessor)	Low	Moderate
Hu et al. [12]	Moderate	Moderate	Low	Moderate (treatment details partially provided)	Low	Moderate (not blinded assess)	Low	Moderate

**Table 3 medicina-57-00783-t003:** Summary of the results of the included studies (forced eruption duration and root measurements).

Authors	Average Forced Eruption Duration (Months)	Data before Treatment (T0)	Root Length (mm) and Resorption (after Treatment)	Conclusions
Chaushu et al. [14]	10 (range 3–12)	Data not presented	Not evaluated	-
Ho and Liao, [16]	8.0 ± 4.5 (range 2.0–24.1)	Data not presented	ICI values: Dilaceration group Root length 7.5 ± 2.5 *. Root resorption: <3 mm, 2% (13); 3–5 mm, 5% (31); >5 mm, 9% (56). No dilaceration group Root length 12.2 ± 3.0 *. Root resorption: <3 mm, 44% (69); 3–5 mm, 16% (25); >5 mm, 4% (6). CCI values: Dilaceration group Root length 12.8 ± 2.9 *. No dilaceration group Root length 14.0 ± 2.9 *.	ICI has significantly greater root resorption compared with naturally erupted CCI. Root resorption correlated with highly and deeply ICI, longer treatment, and root dilacerations.
Shi et al. [18]	10.16 ± 2.73 (range 6.1–14.9)	ICI values: Root length 6.67 ± 1.94 *. CCI values: Root length 9.02 ± 2.13 *.	ICI values: Root length 10.66 ± 2.10. CCI values: Root length 11.04 ± 1.76.	The root length of immature ICI was statistically significantly shorter than that of the CCI at pretreatment. The mean post-treatment root length of CCI and ICI was not significant.
Sun et al. [19]	Not reported	Not evaluated	ICI values: ET group Root length 8.78 ± 1.94 *. LT group Root length 8.39 ± 1.21 *. CCI values: ET group Root length 10.14 ± 2.01 *. LT group Root length 10.75 ± 0.60 *.	The root length was statistically significantly shorter for ICI in comparison to the CCI group. The results of root length are better in the ET when compared with the LT group
Hu et al. [12]	14.41 ± 4.03 (range 6.93–21.03)	Not evaluated	ICI values: DRL (T1) 8.37 ± 1.74. DRL (T2) 10.99 ± 1.96. DIRL (T1) 7.53 ± 0.57 *. DIRL (T2) 9.61 ± 1.69 *. CCI values: DRL (T1) 9.88 ± 1.65. DRL (T2) 11.65 ±1.37. DIRL (T1) 9.88 ± 1.65 *. DIRL (T2) 11.65 ± 1.37 *.	In the follow-up, the root lengths of ICI and the CCI were significantly longer than at post-treatment. The direct dilacerated root length remained shorter.

ICI, impacted central incisor; CCI, contralateral central incisor; DT, discontinuous traction; CT, continuous traction; ET, early-treated; LT, late-treated; DRL, dilacerated root length; DIRL, directed root length; * statistically significant difference between ICC and CCI groups.

**Table 4 medicina-57-00783-t004:** Summary of the results of the included studies (alveolar bone and periodontal measurements).

Authors	Alveolar Bone (mm) (after Treatment)	Periodontal Evaluation (after Treatment)	Conclusions
Becker et al. [11]	ICI values: alveolar bone support (m) 78.5% *. alveolar bone support (d) 79.9% *. CCI values: alveolar bone support (m) 84.6% *. alveolar bone support (d) 84.3% *.	Probing depth ICI values: MLP 2.76; MP 2; DLP 2.94 *; MPP 2.42; PP 2.28 *; DPP 2.34. Probing depth CCI values: MLP 2.52; MP 1.8; DLP 2.63 *; MPP 2.52; PP 1.78 *, DPP 2.26.	A statistically significant difference between ICI and CCI in: -the mean probing depth, especially in the distolabial and palatal areas; -the reduction of alveolar bone support (mesial and distal aspects).
Chaushu et al. [14]	ICI values: alveolar bone support (m) 71.4% *. alveolar bone support (d) 83.2%. CCI values: alveolar bone support (m) 81.6% *. alveolar bone support (d) 83.9%.	Probing depth ICI values: MLP 3 *; MP 1.9; DLP 2.59; MPP 2.59; PP 1.82; DPP 2.2. Probing depth CCI values: MLP 2.32 *; MP 1.72; DLP 2.45; MPP 2.15; PP 1.81; DPP 2.09.	A statistically significant difference between ICI and CCI in: -the mean probing depth, especially in the mesio-labial aspect; -the reduction in the width of the attached gingiva; -the reduction of alveolar bone support on the mesial aspects.
Shi et al. [18]	ICI values: Bone loss labially 2.91 ± 1.63 *; Bone loss palatally 0.93 ± 1.00; The bone thickness of labial alveolar crest 0.73 ± 0.19; The bone thickness of lingual alveolar crest 1.40 ± 0.64 *; Bone thickness at the root apex labially 4.24 ± 1.97; Bone thickness at the root apex palatally 7.17 ± 2.01. CCI values: Bone loss labially 1.40 ± 0.91 *; Bone loss palatally 0.77 ± 0.68; The bone thickness of labial alveolar crest 0.73 ± 0.19; The bone thickness of lingual alveolar crest 1.20 ± 0.41 *; Bone thickness at the root apex labially 4.95 ± 1.41; Bone thickness at the root apex palatally 6.94 ± 1.32.	Not evaluated	The labial bone thicknesses at the alveolar crest and apex of ICI and CCI were significantly thinner after treatment than the corresponding lingual values. Alveolar bone loss on the labial side of ICI increased significantly compared with the CCI, whereas the lingual values did not differ.
Sun et al. [19]	ICI values: ET group: Bone loss labially 2.14 ± 1.22 *; Bone loss palatally 1.72 ± 1.19; Bone thickness labially 2.19 ± 1.15; Bone thickness palatally 7.09 ± 1.02. LT group: Bone loss labially 3.30 ± 1.18 *; Bone loss palatally 2.33 ± 1.41; Bone thickness labially 1.61 ± 1.93; Bone thickness palatally 8.00 ± 1.65.	Not evaluated	The results of alveolar bone loss on the labial side are better in the ET when compared with the LT group
Hu et al. [12]	ICI values: (T1): Bone loss labially 3.20 ± 1.76; Bone loss palatally 2.39 ± 0.95; The bone thickness of labial alveolar crest 1.15 ± 0.49 *; The bone thickness of lingual alveolar crest 2.49 ± 1.47 *; Bone thickness at the root apex labially 2.34 ± 1.66 *; Bone thickness at the root apex palatally 7.45 ± 1.23. (T2): Bone loss labially 3.07 ± 2.32; Bone loss palatally 2.11 ± 1.02; * The bone thickness of labial alveolar crest 0.95 ± 0.31; The bone thickness of lingual alveolar crest 2.31 ± 1.63; Bone thickness at the root apex labially 3.73 ± 2.57; Bone thickness at the root apex palatally 7.12 ± 1.72. CCI values: (T1): Bone loss labially 1.66 ± 0.42; Bone loss palatally 1.14 ± 0.63; The bone thickness of labial alveolar crest 1.06 ± 0.33 *; The bone thickness of lingual alveolar crest 1.63 ± 0.58; Bone thickness at the root apex labially 5.05 ± 1.13 *; Bone thickness at the root apex palatally 6.41 ± 0.87 *. (T2): Bone loss labially 1.81 ± 0.63; Bone loss palatally 1.39 ± 0.39 *; The bone thickness of labial alveolar crest 0.79 ± 0.23; The bone thickness of lingual alveolar crest 1.57 ± 0.40; Bone thickness at the root apex labially 3.60 ± 1.75; Bone thickness at the root apex palatally 7.27 ± 1.48.	Not evaluated	The lingual alveolar bone loss of ICC was greater than that of CCI immediately after treatment, whereas the labial losses did not differ. The labial bone thickness at the apex of the impacted incisors increased significantly between T1 and T2.
Sfeir et al. [20]	Diff ICI- CCI values: DT group— MBL 0.19 ± 0.4. LBL 0.08 ± 0.13 **. DBL 0.29 ± 0.56. PBL 0.09 ± 0.13 **. CT group— MBL 0.2 ± 0.24. LBL 0.45 ± 0.12 **. DBL 0.23 ± 0.21. PBL 0.38 ± 0.13 **.	Probing depth Diff ICI- CCI values: DT group— MP 0.19 ± 0.31 **; LP 0.35 ± 0.30; DP 0.23 ± 0.25; PP 0.27 ± 0.25. CT group— MP 0.43 ± 0.31 **; LP 0.43 ± 0.13; DP 0.4 ± 0.33; PP 0.23 ± 0.36.	DT and CT groups showed statistically significant difference for the following measurements: -Mesial probing (MP); -labial bone level (LBL); -palatal bone level (PBL). The use of a discontinuous Ortho-T technique can provide better results in periodontal status and a net reduction in bone height loss.

ICI, impacted central incisor; CCI, contralateral central incisor; DT, discontinuous traction; CT, continuous traction; MP, medial probing depth; DP, distal probing depth; LP, labial probing depth; PP, palatal probing depth (mm); MLP, mesio-labial probing depth; DLP, disto-labial probing depth; MPP, mesio-palatal probing depth; DPP, disto-palatal probing depth (mm); MBL, mesial bone level; LBL, labial bone level; PBL, palatal bone level; DBL, distal bone level (mm); Diff, mean differences in measurements between ICI and CCI. * statistically significant difference between ICC and CCI groups. ** statistically significant difference between DT and CT groups.

## Data Availability

Not applicable.

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
