# Peer review of "Post-Treatment Status of Impacted Maxillary Central Incisors following Surgical-Orthodontic Treatment: A Systematic Review"

_medicina, 2021, doi:10.3390/medicina57080783_

Round 1
Reviewer 1 Report
The authors' systematic review aims to investigate the effects of surgical-orthodontic treatment on impacted maxillary incisors. The topic is of great clinical interest. The study is well realized, the methodology is consistent and the presentation of the results linear. I appreciated the partition of the discussion into subsections. I suggest to the authors some small changes to improve the quality of the manuscript:
_ Line 38-41: Reference 2, specify that this is a retrospective study and that the incidence of 2.0% concerns a sample of Chinese population.
_ I suggest implementing the introduction by specifying better the causes of the incisors’ inclusion. A possible study that the authors could consider is the following (but also other publications if they deem it more appropriate): Chaushu, S., Becker, T., & Becker, A. (2015). Impacted central incisors: factors affecting prognosis and treatment duration. American Journal of Orthodontics and Dentofacial Orthopedics, 147 (3), 355-362.
_ Line 88-89: were the studies regarding cases of malformation or dental anomalies of the upper incisors also excluded?
_ I suggest improving the presentation of the tables. Each table spans too many pages.
Reviewer 2 Report
Manuscript ID medicina-1315507
„ Post-treatment status of impacted maxillary central incisors following surgical - orthodontic treatment. A systematic review“ by Žarovienė et al
The prevalence of impacted teeth ranges from 1% to 3.5% in the general population. Mandibular third molars are the most common impacted teeth, followed by maxillary third molars, maxillary canines, mandibular canines, premolars and maxillary incisors.
Although retrospective analyses showed that spontaneous eruption of impacted incisors after surgical removal of supernumerary teeth is more likely at a younger age
However, a number of impacted incisors (30- 54%) will require further surgical/orthodontic intervention. Evidence-based markers to aid this assessment are, however, lacking.
Therapy then usually involves surgical exposure of the impacted tooth and the application of traction forces via an attachment to the impacted tooth in either an open or closed eruption procedure. The success of such a combined treatment is usually excellent. In this context, the closed technique may have advantages with regard to the crown length and bone availability later on.
However, final evidence is lacking. Thus, the present review, in principle, deals with an interesting topic. Unfortunately, however, it must be noted that the availability of primary evidence is limited and it was to be expected that such a project would not lead to a clear conclusion. However, this is of course not the fault of the authors.
The clinical question was generated using the PICO process and its protocol pre-registered at PROSPERO. It is stated that the review was conducted in accordance with PRISMA recommendations. The authors conducted a search in three electronic database and one registry of clinical trials (search terms and flow chart are provided) and selected articles describing randomized controlled studies, nonrandomized controlled and other trials, excluding case reports/ series, reviews and animal studies. Study selection and data extraction were performed in duplicate and the included studies were described in adequate detail. Quality assessment was performed using Cochrane’s risk of bias tool or the ROBINS-I tool, respectively. The authors declare no competing interests and received no external funding.
Overall the search strategy is acceptable, however, according to the stated guidelines
- the search should include consulting content experts in the field and grey literature.
- a list of excluded studies should be provided
- funding sources of the included studies should be reported
- possible heterogeneity (preventing meta-analysis?) of the studies should be further explained or discussed.
The authors should add the relevant information or at least comment on these points.
The topic of manuscript is of general interest and the review mostly comprehensively written. This makes it all the more that some elements of a critical review are missing.
Overall, the manuscript, in its present form is not suitable for publication. The authors should strongly consider adding the indication of evidence levels and the specific points listed above. I would therefore ask the authors to revise the manuscript according to the suggestions and to resubmit it in revised form.
Author Response
Author's Response
We thank the reviewers for their valuable input, which contributes strongly to improve the content.
Reviewer 2
Overall the search strategy is acceptable, however, according to the stated guidelines the search should include consulting content experts in the field and grey literature.
Yes, we agree, that grey literature could provide additional information. However, before preparing the systemic review, the authors decided to exclude grey literature due to the risk to add low-quality work.
- a list of excluded studies should be provided
Please kindly find the attached file with excluded studies.
- funding sources of the included studies should be reported
Two out of 7 studies were supported by grants (12, 19).
- possible heterogeneity (preventing meta-analysis?) of the studies should be further explained or discussed.
Clinical heterogeneity was high between studies in terms of the diversity between the age of participants and outcomes measurement, which makes meta-analysis impossible.
We thank the reviewer for the comment.

Reviewer 3 Report
I would like to congratulate the authors for this systematic review focused on status of maxillary central incisors after surgical treatment. I think this kind of papers are very interesting for readers and practicioners.
The paper was prepared according to all scientific requirements for the systematic review and I will suggest to accept it in present form.
